# A Review of the Role of the School Spatial Environment in Promoting the Visual Health of Minors

**DOI:** 10.3390/ijerph20021006

**Published:** 2023-01-05

**Authors:** Huihui Zhou, Xiaoxia Bai

**Affiliations:** 1School of Architecture and Urban Planning, Huazhong University of Science and Technology, Wuhan 430074, China; 2Hubei Engineering and Technology Research Center of Urbanization, Huazhong University of Science and Technology, Wuhan 430074, China

**Keywords:** health promotion, learning environment, myopia, outdoor exposure, school spatial environment, visual health

## Abstract

Rising childhood myopia rate has detrimental health consequences that pose a considerable challenge to health systems. The school spatial environment, which is where students are for the longest period of time, has a high health value for myopia systematic intervention. While research has demonstrated associations between physical daylight environments, medical gene and visual health, the literature currently lacks a synthesis of evidence that will act as a spatially-organized resource for school designers. This study is based on literature from the period 2000–2022 and has been taken from the Web of Science, scopus, Medline and CNKI core collection database. Collaboration, literature co-citation and quantitative and qualitative analysis, in addition to keyword co-occurrence are adopted to conduct a visual health research review. The results indicate that intensive near work activity (as a risk factor) and longer time spent outdoors (as a protective factor), are involved in visual health factors. Two main research themes are obtained and relate to: (1) The environment of visual work behavior (especially the near work learning environment) and adaptable multimedia learning environment; and (2) the environment of outdoor exposure behavior. Furthermore, with the variation of educational demands, models and concepts, there are different demands for near work behavior, and this study makes an important contribution by pointing to two future research directions, including the accurate and controllable environment of near work behavior, which operate in accordance with various educational mode requirements and the active design of the environment of outdoor exposure behavior. In referring to differences between regions and countries, as well as the development of the educational environment, it provides insight into how these demands can be controlled.

## 1. Introduction

Vision, the most dominant sense, plays a crucial role in obtaining information. Visual health, mental health and levels of well-being are closely interconnected [1], and the human body needs healthy vision in order to function optimally [2]. Rising childhood myopia problems with high incidence that occur in young ages pose a considerable challenge to health systems globally, and this is especially true in Asia [3]. Estimates of growth in the urbanization and human development index suggest that, by 2030, there may be 3.36 billion people with myopia across the globe. The number of people with high myopia, which is frequently accompanied by serious complications, is estimated to increase to 516.7 million in the same year [4]. The COVID-19 pandemic also greatly changed lifestyles, longtime online working and learning in ways that will aggravate the severity of the myopia problem. Take China as an example. As a result of increased large-scale online teaching, decreased physical activities and outdoor exposure during the 2020 pandemic, minors’ visual health levels decreased significantly, and the myopia rate increased by 11 percent [5]. This phenomenon is almost a natural experiment that verifies that near work mode, content and environment directly impact visual health.

As an irreversible process, myopia develops most rapidly in minors since the minors’ eyeball structure is not yet mature and school learning requires heavy near work. The known related factors of minors’ visual health include genetics, environment, diet, sleep, personal habits, educational pressure and so on [6]. Of these factors, environment is related to strong intrusive characteristics, which mainly involve the family and school environment. In contrast to the diversity of family environments, school is a place where minors spend the longest amount of time for near work, and this environment has great value for group interventions that seek to promote visual health. In addition, spatial environment intervention design provides minors with a healthy environment, but this is also the countries’ fundamental responsibility. School spatial environment refers to indoor spaces and outdoor spaces, such as classroom, sports fields, greenery and other spaces, and encompasses both architectural composition and spatial layout. It mainly studies the spatial and environmental characteristics that are related to student behavior [7].

In recent years, visual health research has become more mature, and the relationship between school spatial environment and visual health has gradually been highlighted by pure physical illumination research that increasingly focuses on dynamic and cross comprehensive interventions. Visual health research cuts across multiple fields and it is essential to synthesize the multidisciplinary crossover results for practice. In the review process, studies address minors’ visual health in the school environment, and do so in the expectation that conclusions may be transferred to design guidelines. This study begins by searching for visual health risk factors, and then collects school environmental elements and case studies that focus on visual health, and then uses them to design guidelines.

This paper seeks to bridge the translational gap between multidisciplinary research and school design strategies. This may contribute to designers in practice and enable minors to adopt healthier behaviors that promote visual health. 

## 2. Method

### 2.1. Literature Search

We conducted a comprehensive literature search (through May 2022) that addressed myopia, the school spatial environment and school design, and minors. This study seeks to cover material that could impact the development of a conversion tool that will sustain both architecture designers and researchers who wish to establish an evidence-based visual health-promoting school design; it does not, to this extent, seek to determine or quantify a relationship between a pair of deliberately defined and measured variables.

We searched the following databases: Web of science, CNKI, Scopus and Medline (via pubmed). We used Medical Subject Headings (MeSH) search codes where possible, and used the following terms: (Schools[mesh] OR school*) AND (“facility design and construction” [mesh] OR “environment design” [mesh] OR architecture OR environment OR “school design” OR “building design” OR “built environment” OR “classroom design” OR “school room design”) AND (visual health[mesh] OR myopia/prevention and control[mesh] OR “visual health promotion”[mesh] OR “short-sightedness”[mesh]). In engaging with databases that do not use MeSH, we used a similar key word structure. It should be pointed out that we also searched for these words in Chinese, and CNKI was our choice as the database because it has the most complete amount of Chinese journal data. In particular, some case studies on built cases will enlighten the translation of strategies.

### 2.2. Selection Criteria

In considering the myopia problem, we note that there are already several mapping reviews in both English and Chinese [6,8,9]. Accordingly, when addressing the specific topic of school spatial environment in promoting visual health, we opt for a scoping review methodology that includes different stages [10], which is adapted to translational purposes in practice.

Our review focuses on two themes: “factors related to minors’ visual health; the relationship between school spatial environment and minors‘ visual health”; and “the contribution the spatial environment could make to visual health”. Correlation with these two themes is the basis on which the literature is filtered.

First, thinking about the complexity of the visual system, which is composed of eye, optic nerves and pathways to and between different structures in the brain, means that discussions of minors’ visual health mainly focus on “no refractive errors” (which means not contracting myopia). The literature is not limited to the study of myopia, as environmental factors that promote visual health behaviors are also covered. The reason for this is that the scope of “health” is much wider than “not sick”, which refers to being in a perfect state [11]. Environmental effect is not a short-term sensitive pathogenic factor, but is instead long-term and wide-ranging. This study therefore firstly mapped the position of the environment factor with all other known factors and took this as the basic cognition.

Secondly, we identified 555 unique sources (491 in English and 64 in Chinese) that were potentially relevant to the topic of school spatial environment, and which could be used to promote visual health. This screening and removal of duplicates left 533 items. We selected studies for appraisal in a 2-stage process. First, we scanned titles and abstracts identified from the search strategy and limited articles published in the period January 2000–May 2022. Eligible studies (1) are randomized controlled trials with participants aged 6 to 18 years. This is taken as the baseline as this age range largely covers the age range for primary and middle school; (2) we report factors including visual health internal influence mechanism; and (3) examine related spatial environmental variables as part of outcomes. We exclude studies that did not sufficiently focus on any aspect of child/youth population, spatial environment, myopia or visual health. The only exception we made in this regard is if the work pertained to specific environmental variables or related issues in instances where it was not possible to focus on children’s myopia and school spatial environment myopia interventions. 

Of the 53 full-text sources assessed, 14 were retained for qualitative review. In order to enable translation to the school spatial environment intervention, we focused on 39 sources that are empirical studies or reviews of empirical work, and which pertained to school environmental variables that could potentially be used by architecture designers (Figure 1).

### 2.3. Data Extraction and Analysis

We qualitatively analyzed literature sources to identify study types and designs, sample characteristics, approaches and measures, and key findings. We then engaged in an iterative process of summarizing and synthesizing the findings, assessing relative strengths of evidence in order to construct the role of school spatial environment among visual health sufficiently, and considering how we might best translate evidence to a structure that would be of practical use to school designers—this would then help designers to undertake school design that would promote visual health. 

We rated individual studies’ strength of evidence by referring to research designs and the relationship with the school spatial environment by referring to 3 levels, specifically strong, moderate and preliminary.

Strong evidence came from longitudinal cluster randomized or cluster matched controlled trials that were conducted in school environments. These contained obvious built environmental elements and minor behaviors and could be directly oriented to architectural design.Moderate evidence came from longitudinal approaches with smaller, single-site samples and a comparison or control group. These included placing the surrounding environment or behavior in a social context, and this served to indirectly orientate architectural design.Preliminary evidence first came from case studies related to visual health and then also from medical or genetic research where the associated pathogenic factors are difficult to regulate or modify—this played an indirect but fundamental role in guiding architecture design.

Correlates and causal factors of visual health engaged by these studies were wide-ranging, and in a few cases produced inconsistent results. We therefore discuss the strength of evidence for the identified school spatial environmental variables by referring to their overall support in applicable studies.

Once the relative strength of the evidence is assessed, we re-conceptualized these relevant variables into spatially-oriented design domains that were developed by referring to designers’ work and decision processes. In undertaking this work, we consider core principles that apply when empirical research does not explicitly or concretely inform needed design knowledge. We also design best practice and theory-based pathways to impact that can be considered as testable hypotheses.

## 3. Results

### 3.1. Characteristics of the Included Articles

Many factors may lead to myopia. We identified 53 empirical studies and literature reviews that addressed aspects related to school spatial environment and minors’ visual health—these consisted of 32 (60%) longitudinal designs, 10 (18%) cross-sectional studies, 14 (26%) reviews and one case study (1.8%).

Of the longitudinal measures, six are cluster randomized trials, five are controlled studies, and 24 consisted of within-subject comparisons without an optimized or longitudinal control group. Longitudinal study sample sizes ranged from 102 to 33,355 individuals. Twenty-one studies of self or parent-completed questionnaires were subject to meta-analyses, and this made it possible to construct the independent and explanatory built environmental variables. Visual health is objectively measured with an instrumental method in 18 studies that include several types of refraction with cycloplegic and ocular biometry, cycloplegic autorefraction and axial length and clinical eye examination. 

All cross-sectional studies used quantitative and applied statistical methods to explore related visual health environmental factors. Of these studies, 10 explored behavioral factors (including outdoor activity and near work) and seven studied family factors like living type and location and eating habits. Cross-sectional study sample sizes ranged from 264 to 43,771 individuals.

In the research, 25 studies (64%) took place in China, and the remainder in Canada (1), Delhi (2), Danish (1), America (2), India (3), Singapore (3), France (1), Korea (3) and other countries. Thirty-four studies (86%) evaluated environment factors and 11 (29%) referred to work and outdoor related behavior. The majority had a follow-up of six months or longer.

A total of 11 (79%) reviews focused on the relationship between environmental factors and visual health and addressed factors that included local population density and light environment. Two (14%) examined atropine, and the same percentage of articles (15%) examined genetic and environmental factors in relation to visual health. Most records are narrative reviews (58%), and only a few performed systematic meta-analysis (8%). The review elaborated the influence of genes and environment on visual health in detail, and six focused on the light environment.

Table 1 and Table 2 show the relevant research factors of visual health, the study setting location and population characteristics; in addition, it also provides a research description and provides the selected studies’ outcome measures.

### 3.2. Description of the Visual Health Loop

In order to better understand the relationship between school spatial environment and minors’ visual health, this study draws on established research to construct a loop diagram that describes the inner mechanism.

Ever since myopia research began, several decades of research have addressed genetic inheritance; however, the related population genes have not changed significantly. Meanwhile, the number of people with myopia has increased greatly as a result of changes in the physical living environment and lifestyle in recent years, and this indicates that environment may play a decisive role [33,40] (p. 8). Myopia is not only affected by genetic factors, but also by the biological hormone regulation of the internal physiological mechanism [62,63] (p. 13). 

Furthermore, there is a close relationship between human behavior under the influence of the external environment and biological reaction. For instance, outdoor exposure could boost the secretion of dopamine in minor retinas, and this could regulate eye development and promote visual health [64]. Concurrently, the short-wavelength blue light of sunlight, namely the spectral component of 446–477 nm, is much more abundant in daylight than artificial lighting, and its substance is related to the secretion of melatonin, which can promote circadian rhythm, assist the development of the eyeball, and improve sleep [65]. Behaviors that are subject to different environmental influences will further affect the internal hormone regulation, and this clearly indicates that myopia is a disease caused by the interaction between genetic and environmental factors. Closer inspection shows it is more related to the surrounding environment of minors, after force majeure such as heredity are excluded. Similarly, key influencing factors, such as the minors getting out and enjoying outdoor activities, and the possession of a comfortable near work environment are closely connected to spatial environment characters. Hormones act as the inner core influencing mechanism, and the environment could be artificially designed externally and could influence visual behavior further. In being subject to the influence of the social environment and educational mode, near work behavior and outdoor exposure behavior are the main types of visual health behavior that are related to the spatial environment. Our study aims to provide visual health environment from the perspective of school spatial environment, through the conclusion of biological mechanism research to guide the architecture design. 

These related literatures addressed a broad array of macro to micro-level school environmental characteristics and relationships and considered them in relation to a range of visual health-related measures. In drawing on the literature research, we set the loop picture to describe the ways in which related factors influence visual health (Figure 2).

#### 3.2.1. Medical Related Research

genetic inheritance

(1) Parents myopia

Myopia prevalence is significantly higher among children whose parents both have myopia. The greater the severity of parental myopia, the higher the risk of myopia [23,24,39,43,66,67] (p. 6,8,9). Minors’ eye AL (ocular axial length) values may also have been affected by parental genetics [62]. Parental myopia is also thought to be a symbol of genetic and shared family environmental exposure, and parents with myopia are more likely to create an environment that creates visual fatigue, including more intensive education or less time out-doors [34,63] (p. 8); however, gene-environment interactions are not well understood.

(2) Susceptibility gene

The minor’s gender affects the probability of myopia, which means it is more probable that girls will develop myopia during adolescence [23,24,68] (p. 6). Medical gene studies have found that at least 10 genes are associated with early-onset changes in refractive error [69]. Additionally, some gene mutations can lead to myopia or other refractive error phenotypes [70].

The risk of myopia developing varies in accordance with age—it progresses most rapidly between the ages of six and seven and slows down after the ages of 11 and 12 [23,33,48,71] (p. 6,8,9).

(3) Atropine

With regard to drug treatment related to Optometry, it has been found that antimuscarinic drugs, such as the most widely studied atropine, slow the progression of myopia; however, some minors do not respond to it, and there has been no large-scale popularization or application [72]. 

peripheral refractive error

(1) peripheral refraction

The central refractive error is determined by the foveal region on the visual axis, although peripheral retinal and other regions also play an important role in eye growth [73]. The degree of peripheral hyperopia varies among different heritage groups [74,75], and minors with myopia have greater peripheral relative hyperopia [76]. Both emmetropic and hyperopic children had peripheral relative myopia at all eccentric points [77]. Peripheral hyperopia co-occurs with prolonged myopia and determines the myopia patients’ eyeball shape. In contrast, a longitudinal study found that baseline peripheral refraction did not predict nor influence the development of myopia [78].

(2) physiological hormone regulation

Recent medical studies found that outdoor exposure could promote the secretion of dopamine in minors’ retina and could regulate eye development and promote visual health [79]. Meanwhile, the short-wavelength blue light of sunlight, namely the spectral components of 446–477 nm, is much more abundant in daylight than artificial lighting. Additionally, this substance is related to the secretion of melatonin, which can promote the circadian rhythm and eye development, and improve sleep quality [65].

#### 3.2.2. Environment Related Studies

educational environment

The difference in education level largely determines the difference in the probability and degree of myopia and intelligence test scores [80]. Education level was also positively correlated with AL length, which is an important judging factor for myopia [81,82]. Meanwhile, early intensive education is associated with higher prevalence of myopia than the other factors [83].Education level is usually measured by years of formal education or academic achievement, and is also highly correlated with near work time. Education level may therefore be a substitute factor for proximity work and number among the influencing factors of myopia [34,45,84] (p. 8,9). The relationship between education and myopia may also reflect the common inheritance of intelligence and refraction, which enrich the method of myopia researches.

When the traditional educational mode turns to multimedia teaching, the old instructional design criteria are no longer suitable. School type [32] (p. 8), desk lighting [40] (p. 8), sitting location in classroom, screening time [19,20,32,33] (p. 6,8) and multimedia teaching under daylight conditions [16] (p. 5), will also influence minors’ visual health through near work. Classroom unit spatial design should also be changed to adapt to the current education model, in order to provide more visual health environment.

spatial environment

The development of myopia is also related to geographical location, and urban dwellers have a higher risk of visual impairment and blindness than those in rural areas, including in China [42] (p. 9), [85], Ghana [86] and Australia [44], this may be caused by differences in lifestyle, urbanization, school type and demographic characteristics [22,32,33,34,47,51] (p. 6,8,9). Meantime, different geographical areas have various climacteric characteristics that are related to differences of daylight environment, this can be seen in Istanbul [87], Chongqing, Lasa [88], London, Chicago, Dubai and Bangkok and so on. London, Chicago, Dubai and Bangkok have different fenestration parameters in order to ensure the effective use of daylighting, due to local daylight performance [89]. Outdoor activities in China’s rural area have a weak protective effect on visual health, and this may be because minors in rural areas have a lower genetic susceptibility to myopia; in addition, environmental factors may be an important reason for visual impairment [90]. Additionally, the indoor environment, like plants, can also influence the relief of visual fatigue [27] (p. 7). After controlling for other interfering factors, we observe that family living environment is an influential factor for minors’ myopia [28,29] (p. 7). Living on a lower floor allows more time to go outdoors, which is a visual health protective factor. While there is less research on the relationship between the environment and visual health, physical environment has been found to play an empirical role in the overall mechanism [36] and influence behavioral choices [91], which provides support for architecture design

outdoor environment

Outdoor daylight exposure could protect visual health to a certain extent and delay the development of myopia, and this is because outdoor time is an important protective factor [15,17,21,23,24,46,52,55,92] (p. 5,6,9–11). Minors who spent less time outdoors and more time near work had a higher rate of myopia, and those who spent less time on near work had a lower rate [26,27,35,36,38,93,94,95] (p.7,8,16), which indicated increased outdoor time as a solution for myopia, and provided architecture design direction. Similarly, animal experiments have also found that a less bright environment can increase the probability of myopia [96].

However, the biological mechanism of the association between outdoor exposure and myopia remains unclear. It has been hypothesized that higher outdoor brightness increases field depth and reduces image blur, and that light stimulation induces dopamine production in the retina and adjusts eye growth [95]. It has also been hypothesized that the spectral component of daylight is an important visual health factor [97]. Studies have also identified the non-visual effects of light and Healthy Circadian Lighting, namely that light has an important impact on the human body’s circadian system. Non- visual photoreceptor cells (ipRGC) transmit light signals to the brain’s suprachiasmatic nucleus (SCN) and body clock and regulate the secretion of melatonin and cortisol (which are closely related to circadian rhythm) [98]. Both these explain how outdoor time could protect visual health.

#### 3.2.3. Behavior Related Studies


negative behaviors related to visual health


Near work is usually defined as long or close reading by minors. Children who read continuously within 30 cm for more than 30 min tend to develop myopia [99]. Meanwhile, children who read more than two hours per day and more than two books per week are more likely to be nearsighted [13,30,35,38,49,50,55,99] (p. 5,7–11). It has also been found that the light environment of near work will influence visual health [14] (p. 5). Meanwhile, outdoor activities can protect visual health [100].

Some studies have also found there is no association between close visual work and vision-related health risks [101,102]. Most myopia and near work are cross-sectional and cannot examine the temporal relationship. Those who have myopia may wear glasses and find it more difficult to participate in sports tasks, which results in lower levels of physical activity [103]. Parents also filled in most of the information about minors’ near work and outdoor time, which led to recall bias. In the future, more accurate and standard methods should be used to quantify the near work modifiable variables in near work, such as reading posture, rest during reading and appropriate lighting. These should be studied in order to promote health [104], and this could change e and adjust behavior in accordance with spatial environment design. The increased burden of school learning plays an important role in the high prevalence of myopia and visual impairment. This may be due to intensive near work, which increases the risk of myopia [33,105]. Bad eye habits and electronic devices are also risk factors [106,107]. Family eating habits and BMI risk factors [43] (p. 9), including changing from a Japanese to a Western diet, may increase the risk of myopia [18,19,108] (p. 5). Additionally, these factors could be improved in other ways, like social management policy and media guidance. This can help to produce positive behaviors related to visual health positive behaviors related to visual health.positive behaviors related to visual health

Resting behaviors (including night’s sleep and short rest) have been associated with the probability of minors’ myopia [25,50] (p. 7), that could protect visual health. The level of myopia is significantly associated with night sleep, and not with the duration of total noon and evening sleep [37,54,100] (p. 8,11). With regard to the biological mechanism, the circadian rhythm of human sleep mainly depends on the joint regulation of melatonin and dopamine, when melatonin combines with the corresponding eye receptors to jointly regulate the eye’s development [66] (p. 14). Short rests, meanwhile, have been found to reduce fatigue [103]. Some animal experiments have also demonstrated that the circadian rhythm disorder will increase the risk of myopia [109]. Change of minors’ daily waking and sleeping time is a simple measure that can be introduced to restrain myopia [110]. 

Eye exercises organized by schools have a certain protective effect on myopia [44,111] (p. 9). In addition appropriate myopia correction can also protect visual health, further reduce visual damage and slow myopia progression [112]. Parents’ awareness of refractive error also determines the proportion of minors‘ myopia correction, and this further affects visual health [53,113] (p. 11).

### 3.3. Visual Health Guidelines for School Architecture

In referring to the loop diagram’s logical chain, we see that it is both plausible and likely that the school spatial environment will impact on minors’ visual behavior. However, few review studies have discussed associations with minors’ visual health. We combined the relevant studies and summarized the value of school spatial environment for systematic myopia intervention. In the causal loop diagrams of visual health, school layout is shown to be closely related to the indoor light environment through effects on near work behavior; in addition, spatial environment greatly influences minors’ outdoor times via hormone regulation and physiological processes in daylight. The natural elements could also relieve visual fatigue, and this applies both indoors and outdoors. These studies use direct and indirect routes to combine visual health promotion and school design.

Usually, visual health is always connected to near work behavior that is related to indoor light environment in school. Additionally, school layout, windows’ scale, buildings’ orientation and so on are the main influences. On the basis of the loop diagram logical chain, we therefore propose that light environment changes due to technology also need to be taken into account. For the most part, visual health promotion has not only considered physical school environment variables but has provided general support to providing minors with adequate outdoor activities. For example, diverse playgrounds, convenient outdoor access, various types of activity equipment and varied travel experience all show how this can be achieved. While the specific relationship between school spatial environment and visual health has not been precisely defined, it has been demonstrated that the school environment promotes visual health opportunities. Minors’ outdoors activities at school have decreased dramatically over preceding decades, and this is why the importance of outdoors time as a way of improving visual health has been increasingly reiterated. Unfortunately, many schools and surrounding environments have not been conducive to outdoor activities. In addition, minors who lived in cities were more prone to myopia than rural counterparts, and in this regard it was significant that urban schoolchildren enjoyed substantially less school outdoor time. We identified empirical studies and literature reviews that engaged school spatial environmental design and minors’ visual health. In contrast to many of the reviews that identified advantages and difficulties in the promotion of visual health, including changes to the physical environment, this review was instead focused on the school’s spatial ‘designed’ environment. 

The complex causal pathways between school spatial environmental factors and minors’ visual behaviors are still unclear. However, in acknowledging the ongoing need to improve visual health behaviors across numerous minor populations, schools have introduced and promoted design guidelines (see Table 3). They are summarized on the basis of the literature review and relevance analysis.

## 4. Discussion

(1) Benefits of other dimensions

The universality of the myopia problem and the vital practical significance of our re-search suggests that guidelines could be applied in other areas to promote health. For example, introducing biophilic design elements into the office, such as daylight and greenery, could improve human well-being, performance, creativity and health [133]. In addition, developing healthy circadian lighting for inpatient ward could compensate for the lack of daylight; considering community walking/cycling and public transportation; and encouraging good eating habits could also promote health.

Meanwhile, the role of the school spatial environment certainly has multiple health benefits, and this is because of the complex causal pathways between visual health and human behaviors. For example, outdoor time could increase physical activity and promote optimal functioning [134]. Other studies have shown that it can address other health problems and contribute various health benefits (including promote physical fitness) [135] and improve academic achievement [136,137,138].

(2) Benefits under the life dimension 

The diopter between minors’ hyperopia and emmetropia is regarded as the regulatory reserve. The newborn’s binocular hyperopia gradually tends to be normal along with growth. [139]. If the minors’ eye axis length is prematurely developed to 24 mm, then the adjustment reserve is exhausted in advance, and they are more likely to develop myopia. It has been hypothesized that screen exposure in early life may be related to the occurrence of myopia in preschool minors [140]. Unhealthy minor behavior, including excessive reliance on electronic screens, spending too long in the artificial lighting environment and limited outdoor activities are leading to premature minor consumed hyperopia reserves and even myopia. The early detection of hyperopia reserve deficiency and timely intervention both have a key role to play in protecting minors’ vision [141]. Different architectural types could link all the life stages, and this could help to control the progress of myopia, which means the prevention and control of myopia can be classified into different targets based on stages of visual health. For instance, it is possible to maintain visual health situation, and delay the myopia process after getting myopia. It is of the utmost practical importance to design architectural spatial environment that promotes visual health.

In the myopic loop diagram, the visual health mechanism is diversified. Our review only discusses the impact of the school spatial environment on visual health, and it does not therefore engage other factors, such as social management and medical treatment that could benefit the protection of visual health. 

(3) Benefits of society dimension

As architectural designers, we should consider the surrounding environment and the housing type with the aim of providing a visually healthier living space [142]. Of course, it is also essential to take into account other family-related influencing factors, like eating habits and transportation modes. Additionally, social constructs that could impact gender differences also require more discussion and exploration [24].

There are substantial differences in education modes across different regions and countries, and this includes America’s outdoor campus with Richard Noitra’s natural school. However, Chinese research of myopia and the construction of educational buildings is still at a relatively early stage, and is still, taking into account the daylight environment, concerned with traditional forms and structures. At the same time, the construction of short time with high population density has greatly influenced the Chinese educational mode, with result that the majority are enrolled in large classes and that minors are mostly taught in fixed classrooms and encouraged to stay indoors during breaks. This contrasts with the walking classes of America, which promote activity and increased daylight exposure between classes. This is just one point of possible reference for the future development of the Chinese educational model. Others should be considered to maximize health promotion.

## 5. Conclusions

This study engages with the school spatial environment design process to show how changes in school spatial environments can promote visual health. It then builds on this insight to put forward feasible strategies that could be used to guide school design towards desirable visual health outcome. These include the accurate and controllable environment of near work behavior, which operates in accordance with various educational mode requirements and the active design of the environment of outdoor exposure behavior (see Table 3). However, shortcomings in the medical and empirical research evidence base will need to be addressed and overcome if progress is to be made in this regard. In addition, while the guidelines seek to focus school spatial environmental design decisions on minors’ visual health, they do not yet include a “formula” or identify specific design strategies that will meet building codes; in addition, they also currently omit many details, including spatial form and material specifications.

## Figures and Tables

**Figure 1 ijerph-20-01006-f001:**
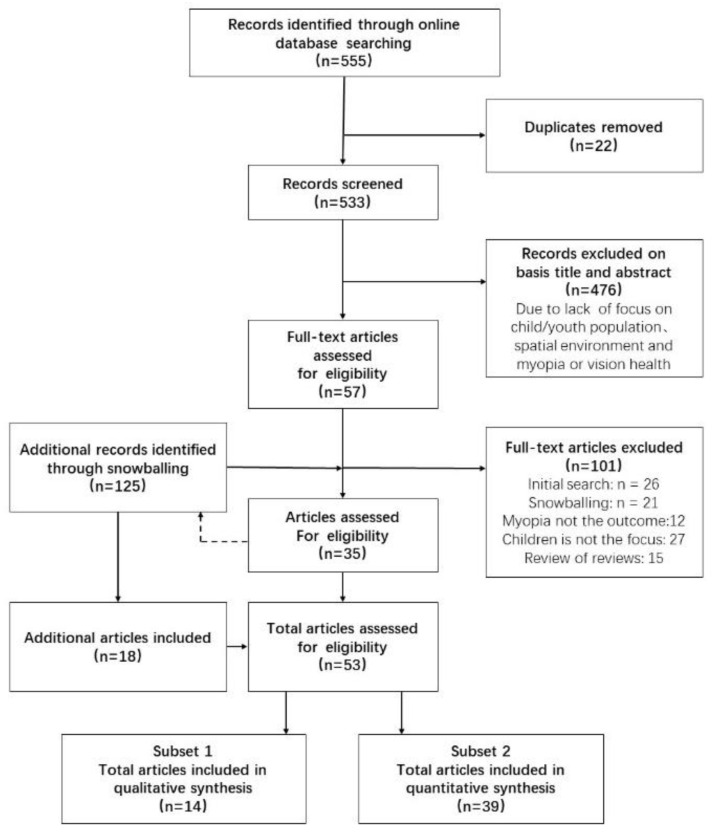
Flowchart: identification and selection of review papers for inclusion in the review. Note: Figure adapted from Libera [12].

**Figure 2 ijerph-20-01006-f002:**
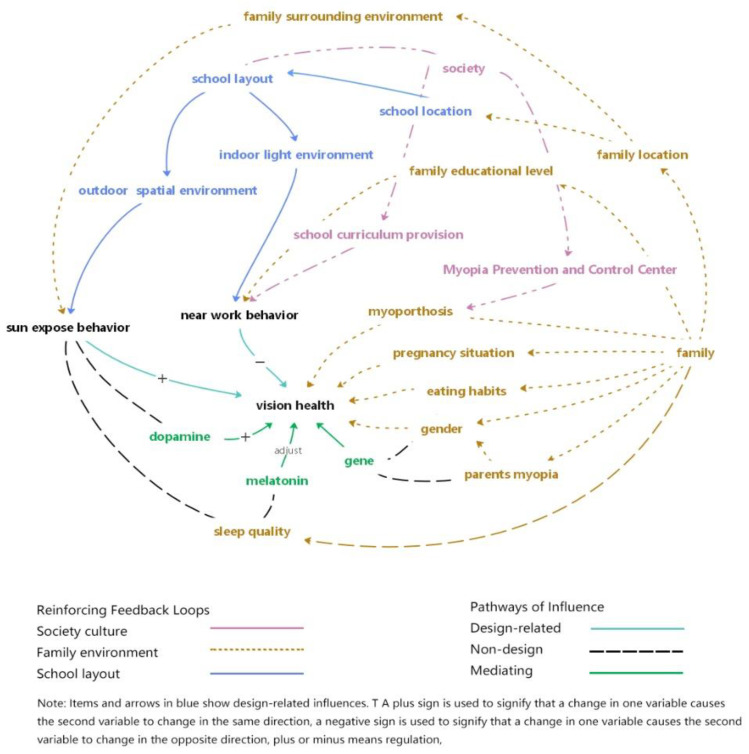
Causal loop diagram of a genetic-environment-behavior system for addressing visual health.

**Table 1 ijerph-20-01006-t001:** Overview of researches in the Review of Visual Health Risk Factors in Minors.

Research Relevant Factors	Study Setting Location	Population Characteristics	Research Aim	Research Description	Research Duration
personal eye habits related myopia [13]	Guangdong province, China.	2289 students online and 2570 students on-site	To examine the prevalence of myopia and its related personal eye habits.	Using stratified random cluster sampling with eye examinations and questionnaires	One month (July 2020)
light levels in classrooms [14]	Northeast of China	317 subjects from 1713 eligible students (6–14) in four schools	To determine whether elevated light levels in classrooms in rural areas can protect visual health.	A stratified cluster sampling visual acuity (VA) test was applied three times (at baseline, 6 months and 1 year after intervention) through eye examinations and a questionnaire.	One year (2011–2012)
family, activity, and school factors [15]	Taiwan, China.	Taiwanese children in Grades 4–6	To explore the effect of family, activity, and school factors on myopia risk and severity.	National cross-sectional data, bivariate and multivariate analyses	Unspecified
multimedia teaching material under naturally varying classroom lighting conditions [16]	Taiwan, China.	children in grades 4–6 across 87 schools	To determine whether exposure to digitally projected and multimedia teaching material under naturally varying lighting conditions is associated with visual health.	A population-based, cross-sectional study	One year
outdoor time, near work and parental myopia. [17]	Beijing, China.	382 grade-1 children (age: 6.3–0.4 years) with 305	To investigate factors associated with ocular axial elongation and myopia progression	Follow-up research and longitudinal designs	Four years follow-up
age, habitation, gender body mass index, body height, school type [18]	Beijing, China.	54 schools randomly selected from 15 districts, including 35,745 samples	To assess prevalence and associated factors of myopia and high myopia in schoolchildren in Greater Beijing.	A school-based, cross-sectional investigation	In 2016
parental myopia, outdoor time, near work, screen time, physical activity, and eating habits. [19]	France	264 children aged 4 to 18 years attending the Centre Hospitalier Universitaire Gui de Chauliac	To evaluate risk factors for paediatric myopia in a contemporary French cohort while takingt eating habits into account.	An epidemiological cross-sectional study	One year (May 2007–May 2008)
age, screen time near work distance, outdoor activity [20]	Bahir Dar city, Northwest Ethiopia,	school children between 6–18 years of age in Bahir Dar city	To assess the prevalence and associated factors of myopia among school children	A school-based cross-sectional study	One month (October 2019–November 2019)
light exposure and physical activity levels [21]	America	102 children (41 myopia and 61 emmetrope) aged 10–15	To objectively assess daily light exposure and physical activity levels in children.	Using a wrist worn actiography device to measure visible light illuminance and physical activity	2-weeks
gender, age, parental myopia, sitting location, near work, and outdoor activities [22]	Nantong, China	At least two classes from each grade of each school in Nantong.	To investigate the prevalence of myopia and the factors associated with it among students in Nantong.	School-based cross-sectional study; randomly selected; self-reported questionnaire	Unspecified
outdoor time, near work, parental myopia and gender [23]	Gongshu District of Hangzhou City, China	1004 third-grade students	To study the prevalence and risk factors of myopia in the initial stage among primary schools, and to provide protective suggestions.	Myopia questionnaires; eyesight-related parameters	three months (November 2017–February 2018)
individual characteristic and eye habits; the influence of social cognition variables on myopia prevention behaviors [24]	Guangdong, China	4894 students at 6 junior high schools in 5 prefectures of Guangdong province	To examine the prevalence of myopia and its related personal eye habits among junior high school students and to explore stage-specific myopia prevention behaviors.	stratified random cluster sampling 2 289 students online and 2 570 students on-site with questionnaire	July 2020
outdoor time [25]	Canada	166 children	To determine the prevalence of myopia, proportion of uncorrected myopia and pertinent environmental factors among children.	Refraction with cycloplegia and ocular biometry were measured in children from two age groups. Parents completed a questionnaire	One year 2014–2015
long-term excessive use eye, outdoor activities and gender [26]	China	Primary students (11,246) junior students (3673) senior students (4220)	To explore the situation and the affect factors of myopia and scientificalness and effectiveness of eye exercises.	Random cluster sampling, A questionnaire was distributed	Unspecified
flat room, living floor and outdoor time [27]	China	43,771 children from 12 cities	To evaluate living environment’s impact on school myopia in Chinese school-aged children.	A large cross-sectional sample of area- and ethnicity-matched school children, questionnaire	March and June 2012
parents myopia, environmental factors [28]	China	353 farmers and 162 farmer families	To quantitatively assess the role of heredity and environmental factors in myopia. The family was referred to in order to establish an environmental and genetic index.	A pedigree analysis with one child (university student), father, and mother; in a multiple regression analysis; 114 pedigree; milies were used as a control group.	Unspecified
near work, social-constructed gender difference on myopia [29].	Yunnan, China	the nationally representative data of CEPS	To verify there exists a difference in myopia prevalence.	Clinical eye examination and questionnaire	In 2014
school type, gender, age, parental myopia, socio-economic status, screening time [30].	Delhi	9884 children (66.8% boys)	To assess prevalence of myopia and identify associated risk factors in urban school children.	A cross-sectional study questionnaire	Unspecified
gender, age, parental myopia, socio-economic status, near work, screening time [31].	Delhi	10,000 school children aged 5–15	To evaluate the incidence and progression of myopia along with the factors associated with the progression of myopia in school going children in Delhi.	Prospective longitudinal study interval screening	One year
sociodemographic factors, (income and education) height [32]	Korean	A total of 33,355 Koreans over a five-year period.	To evaluate the association between myopia and risk factors, including anthropometric parameters.	Korea National Health and Nutrition Examination Survey data	Four years
homework and outdoor time [33]	Baoshan District, Shanghai, China	842 migrant children from 2 migrant schools and 1081 from 2 local schools	To compare patterns of myopia prevalence and progression between migrant and resident children.	Random sample Baseline measurements were taken of children in grades 1–4, and children in grades 1 and 2 were followed for 2 years.	Two years
physical activity [34]	Danish	307 children	To determine associations between physical activity (PA) and myopia in Danish school children and investigate the prevalence of myopia.	A prospective study with longitudinal data. PA was measured objectively by repeated ActiGraph accelerometer measurement four times with different intervals (1–2.5 years).	in 2015 (7-years follow-up)
living area, age, gender, sleep duration, and outdoor time [35].	Shanghai, China	6295 school-aged children	To offer new insights to future myopia aetiology studies as well as aiding the decision-making of myopia prevention strategies.	Follow-up study with eye examination.	Two-years
near work and outdoor activities [36]	Handan rural, China	572 (65.1%) of 878 children (6–18 years of age)	To evaluate the relationship of both near work and outdoor activities with refractive error in rural children in China.	Cross-sectional study Handan Offspring Myopia Study (HOMS).	Three months (March 2010–June 2010)
parental myopia, age and household income [37]	Korean	3862 children from 5–18 years of age from 2344 families	to investigate the effect of parental refractive errors on myopic children in Korean families	The ophthalmologic examination dataset of the Korean National Health and Nutrition Examination Surveys IV and V.	Four years (2008–2012)
content and time of near work, screening time and desk light [38]	Anyang, China	770 grade 7 students with mean age of 12.7 years	To examine the associations of near work related parameters with spherical equivalent refraction and axial length.	Examined with cycloplegic autorefraction and axial length.	Two months (October 2011–December 2011)
Chinese cities location (linked to light exposure), education level, and gender [39]	Northern and Southern China	9171 primary school students (grades from 1 to 6)	To assess the myopia prevalence rate and evaluate the effect of sunshine duration on myopia.	This prospective cross-sectional study National Geomatics Center of China (NGCC) and China Meteorological Administration provided data	Two months (October 2019–November 2019).
age, parental myopia, BMI (Body Mass Index) [40]	South Korea.	983 children 5–18 years of age	To evaluate the prevalence and risk factors associated with myopia and high myopia in children in South Korea.	Korean National Health and Nutrition Examination Survey 2016–2017 (KNHANES VII) data	Unspecified
Chinese eye exercises [41]	Anyang, China	Eligible 201 of 260 children at baseline	To investigate Chinese eye exercises and factors associated with the development of myopia.	Case-control study	Two years (September 2011–November 2013).
higher degree of education (such as attendance of schools, and time spent for indoors versus outdoor time) [42]	Beijing and Shandong, China; India,	3468 adults;(mean age:64.6 ± 9.8 years; range: 50–93 years)	To examine if education-related parameters differ between high myopia in today’s school children and high pathological myopia in the contemporary elderly generation.	The population-based Beijing Eye Study and Central India Eye and Medical Study, and the children and teenager populations of the Shandong Children Eye Study, Gobi Desert Children Eye Study, Beijing Pediatric Eye Study, Beijing Children Eye Study, Beijing High School Teenager Eye Study	Unspecified
outdoor activities [43]	both urban and rural Northeast, China	3051 students of two primary and two junior high schools	To test the impact on myopia development of a school-based intervention program aimed at increasing outdoors time.	Case-control study self-questionnaire and parents questionnaire	One-year
sociodemographic, environmental factors, regional environment differences [44]	Sydney	2367 children and their parents.	To examine associations between myopia and measures of urbanization in a population-based sample of 12-year-old Australian children.	Questionnaire data on sociodemographic and environmental factors from 2367 children (75.0% response) and their parents.	Two years (2003–2005)
age, parental myopia, the serum 25-hydroxyvitamin D concentration, near work and inflammation reflected by white blood cells counts [45]	Korea	3398 subjects aged 19–49	To evaluate the prevalence and risk factors of myopia in adult Korean population.	Korea National Health and Nutrition Examination Survey 2013–2014 (KNHANES VI). Data	Unspecified
outdoor time and time spent indoors [46]	Beijing, China	643 returned for follow-up examination of 681 students	To assess whether a change in myopia related oculometric parameters was associated with indoors and outdoors activity.	One-year follow-up the longitudinal school-based study	one-year
gender, grade, near work and parental myopia [47]	Guangzhou, China	3055 students of grades 1–6 and grades 7–9	To estimate the prevalence of myopia and to explore the factors that potentially contribute to myopia.	Refractive error measurements and a structured questionnaire data	In December 2014
environment, time of near work, heredity, nationality, grade, and outdoor time [48]	Guizhou, China	7272 qualified students of 8413 students	To investigate the prevalence of myopia among urban and rural students in the Xingyi city of Guizhou province and to analyze the influencing factors.	Visual acuity, refractive examination, were examined among all the subjects, and a questionnaire was analyzed by using logistic regression.	Three months (August 2019–November 2019)
genetics, gender, education, parental myopia, onset age of myopia, outdoor activities, vision care knowledge [49]	Taiwan, China	522 schoolchildren with myopia	To understand the risk factors for its development and progression and to identify if they are important to public health.	Observational studies self-questionnaire	Nine months (February 2018–November 2018)
sleep quality [50]	Tokyo, Japan.	486 participants aged from 10–59	To evaluate sleep quality in myopic children and adults.	The Pittsburgh Sleep Quality Index (PSQI) and Hospital Anxiety and Depression Scale (HADS) questionnaire	Three months (January 2014 –March2014)
parental myopia, near work time and outdoor time [51]	Aba, Nigeria.	1197 (male: 538 and female: 659) children 8 and 15 years	To assess the influence of near work, time outdoor and parental myopia on the prevalence of myopia	Myopia risk factor questionnaire cycloplegic refraction	Unspecified

**Table 2 ijerph-20-01006-t002:** Reviews of Visual Health Risk Factors.

Research Relevant Factors	Study Setting, Location	Study	Number of Studies Reviewed	Main Findings	Method of Synthesis
environmental factors [8] (p. 2)	Singapore	Worldwide prevalence and risk factors for myopia	53	The precise biological mechanisms through which the environment influences ocular refraction are a matter of debate. Outdoors time is an important factor in the prevention of myopia.	Systematic review, meta-analysis
exposure to outdoor ambient daylight [52]	Indian	Increasing time in outdoor environment could counteract the rising prevalence of myopia in Indian school-going children	29	The article describes the current myopia scenario in India; identifies ways to tackle the future epidemic. It considers the importance of day light exposure in counteracting myopia and reported possible public health policies for initiation at a school level that could potentially help in myopia prevention and control its progression.	Narrative review
ow-concentration atropine and outdoor time [53]	America	Myopia Control: A Review	53	Antimuscarinic agents include pirenzepine and atropine, low-concentration atropine and outdoor time have been shown to reduce the likelihood of myopia onset.	Narrative review
educational pressure(read for long hours), outdoor time, low-dose atropine and orthokeratology [3]	Asia	Stopping the rise of myopia in Asia	160	More time outdoors and low-dose atropine show best effect in reducing the incidence of myopia and delaying its the onset. Low-dose atropine, orthokeratology, executive prismatic bifocals, and multifocal soft contact lenses have been shown to be remarkably effective in slowing myopia progression.	Narrative review
near work activity, environmental factors, markers for myopia in the human genome [54]	Singapore	Myopia: gene-environment interaction	Unspecified	Both genes and environmental factors may be related to myopia. There are no conclusive studies at present, however, that identify the nature and extent of possible gene-environment interaction.	Narrative review
light intensity outdoors, the chromaticity of daylight or vitamin D levels. [55]	Singapore	A review of environmental risk factors for myopia during early life, childhood and adolescence	72	Population-based data show a consistent protective association between time outdoors and myopia. Evidence for the association of near work with myopia is not as robust as time outdoors and may be difficult to quantify.	Narrative review
educational environment [56]	China	Educational Environment: The Most Powerful Factor for the Onset and Development of Myopia among Students	35	The study identifies the education environment as the primary factor that causes the onset and progression of student myopia, the paper fully recognizes the scientific rationality of and the specific role served by education-medicine synergy in student myopia prevention and control.	Narrative review
near work (time and intensity) and myopia, and its possible mechanisms [10] (p. 2)	China	Relationship between near work and the development of myopia in adolescents	Unspecified	The article summarizes the epidemiological studies on the relationship between near work content, total time spent on near work, intensity of near work and myopia, and its possible mechanisms, and does so with the aim of providing a point of provide reference for myopia epidemiology and etiology.	Narrative review
light environment(electronic light) [57]	Japan	Progress and Control of Myopia by Light Environments	50	Until the ideal pharmacological targets are found, manipulating light environment is the most practical way to prevent myopia. Approximately 2 h of outdoor light exposure per day is recommended.	Narrative review
natural light [58]	Switzerland	Pandemic of childhood myopia. Could new indoor led lighting be part of the solution?	82	Heritability is one of the factors most linked with young myopia together with increased near-distance work, however, there is no evidence that children inherit a myopathic environment or a susceptibility to the effects of near-distance work performed from their parents.	Systematic review
outdoor time [59]	Shanghai, China	Time spent in outdoor activities in relation to myopia prevention and control: a meta-analysis and systematic review	51	To evaluate the evidence for association between outdoor time and (1) risk of onset of myopia (incident/prevalent myopia); (2) risk of a myopic shift in refractive error and (3) risk of progression in myopia.	Systematic review
relationship between gene-environment interaction and myopia [60]	Nanjing, China.	Gene-environment Interaction in Spherical Equivalent and Myopia: An Evidence-based Review	Unspecified	To systematically research association between gene-environment interaction and the myopia/spherical equivalent.	Narrative review
genetic and environmental factors [61]	Denmark	Genetic and environmental effects on myopia development and progression	33	To summarize the relationship between gene-environment interaction and myopia; and to find the interaction effect of the gene or genetic risk score with the environment.	Narrative review
parental myopia, ethnic differences, outdoor time, near work, population density and socioeconomic status [6] (p. 2).	Poland	A review on the epidemiology of myopia in school children worldwide	55	To review the current literature on epidemiology and risk factors for myopia in school children (aged 6–19 years) around the world.	Systematic review

**Table 3 ijerph-20-01006-t003:** Visual health design guidelines for school architecture.

Design Domains	Strategies	Relevant Literature	Evidence Rating
1. Provide precise and controllable environment of visual operation behavior			
Control the stability of the daylight environment			
	Using the windows construction technology	[114,115] [116,117]	★
	Considering the school’s layout to maximize the internal visual daylighting	[118] [119,120]	★
	Using the teaching unit’ s internal structure measures	[121] [122,123]	✰
	Account for site constraints and benefits to optimize the internal visual daylighting	[123,124]	✰
Enhance the comfort of the mixed light environment			
	Arranged light fixtures in accordance with teaching type and the requirement of multimedia screen cooperation	[14] (p. 5)	✰
	Design artificial illumination parameters in accordance with the variation trend of daylight in different climacteric areas	[14] (p. 5), [61] (p. 12)	★
	Consider the photobiological safety of the display screen	[33] (p. 8)	✰
	Consider the Healthy Circadian Lighting on the basis of the non-visual effects of daylighting	[102] (p. 17)	✰
Enhance the resilience of the visual background environment			
	Consider the indoor environment factors’ design,	[27] (p. 7) [124]	✰
	(a) selection of background wall material (b) selection of background wall color (c) selection of background wall’s material and color collocation (d) selection of indoor green configuration		
	Combine daylight and the external landscape to utilize view window to provide visual connection to the outdoors	[122]	○
Improve environmental adaptability for visual work tasks			
	Optimize the scale of teaching unit according to the light spatial distribution and visual distance of multimedia teaching	[122,123]	✰
	Consider the sharpness requirement of the multimedia display screen	[16] (p. 5)	✰
	The internal configuration of teaching unit space adapts to the update of education mode	[32,96,125,126] (p. 8)	★
2 Active design for environment of outdoor exposure behavior			
Enhance the interactivity of the outdoor environment			
	Enhance the diversification and interesting of areas near classroom	[127] [128]	✰
	Enhance the diversification and interesting of activity areas	[117,129] [122]	✰ ✰
	Consider the merging of different spatial levels to enhance openness and transparency	[121]	○
Enhance the accessibility of outdoor spaces			
	Improve the land utilization of school design to improve outdoor activity needs’ configuration	[130] [124]	✰
	Provide convenient access for minors to outdoor areas to optimize accessibility	[131] [122]	★
	Consider the influence of school location on minors’ public transportation access	[129]	✰
Improve the diversity of outdoor functions			
	Consider diversified school outdoor spaces design	[117,121]	✰
	Consider minors’ preference to decorate outdoor activity areas	[132] [117]	★
Improve the adaptability of outdoor facilities			
	Consider indoor and outdoor activity places to suit minors’ age and behavior patterns	[117]	○
	Make interactive landscape experiences more themed and flat	[117]	○

Evidence Rating Key: ★ Substantial Evidence = longitudinal cluster randomized or cluster matched studies directly oriented to architectural design strategy and visual health. ✰ Emerging Evidence = longitudinal approaches with smaller, single-site samples and a comparison or control group supporting the strategy exists, but is indirectly orientate architectural design. ○ Best Practice = theoretical support and/or practice-based experiential support for the strategy, but based on circumstantial evidence.

## Data Availability

Not applicable.

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
