# Peer review of "A Review of the Role of the School Spatial Environment in Promoting the Visual Health of Minors"

_ijerph, 2023, doi:10.3390/ijerph20021006_

Round 1

Reviewer 1 Report

Relevant article with a current theme and a broad approach, it brings in a text with textual cohesion and coherence a new look at the theme, outlining perspectives for future studies.

Reviewer 2 Report

The article aims at a very interesting subject and its starts as a specific literature review regarding causes for children’s vision health and myopia. The study intends to understand the links between vision health issues and educational space (built or not built). The study is comprehensive and points out lots of elements that contribute to vision problems but in my opinion misses the declared purpose. It is just a good literature review that is more focused on effect more that the cause. The link with architecture design dedicated for educational spaces and connected environment is weak and no clear conclusions are extracted, only generic ones that are kind of obvious for an architect. The link with different geographical and climacteric areas is mentioned but not proper analyzed, with references of special characteristic that are relevant for different climates. Some case studies could help in this area as well. The conclusions state that this article does not give a formula but it should have some clear conclusions that are stated in this part. Final comment regards some mistakes that could be found in the text, some misspells and the information from Table 3 which is not clear what it represents, from where it comes structured like this and the symbols are not explained, it is just a review of elements that connect with references of space attributes and health benefits or issues for this type of illness.  

Reviewer 3 Report

While the topic of the article is interesting, I suggest you reconsider retyping the whole article as it is difficult to follow. In my opinion, it is not fit for review.

The article analyzes more the stages followed in the selection of the articles than highlighting the results. The main problem is that it is not exactly understood if the landscape architects will consider these design criteria for the future and if the green space components affect myopia. 

Sure, studies show that exposure to the light of gadgets and spending several hours in front of screens affect vision, but the title refers more to the space in which they spend their hours, the external and internal environment, and these points are less discussed in the research.

Reviewer 4 Report

Dear authors,

Thank you for your insightful research, it is both an important topic and a compelling read. allow me to add some comments that may further the research:

* In the title, you state that this is a "smart review", now typically this is not a widely known type of review, where you map the existing literature found on the different databases and analyze them, if possible, add a short description of why you call it a "smart review" to the methodology section.

* In Table 3 "Visual health design guidelines for school architecture", please add a key to the meaning of the different characters used to rate the evidence as well as a relevance analysis metric on how you categorized them (meaning how did you come up with the various rankings), as this table is essence of the entire paper, more clarification needs to be put into it. (it is not clear how you reached the rating in the table).

* Also Table 3:

in the "Enhance the resilience of environment of the vision background environment", each of the environmental factors can have a tab of its own with its own significance, (they do not have to be combined to affect vision and eyesight). 

*There are several points that are combined in table 3 that may be sections of their own, (for example the enhancement of both the hard and soft surfaces in the outdoor environment, each may have a direct effect on the visual experience whether alone or combined)

* You also need to correlate between the different design domains that you found in your review, including the different design strategies and those that you added in your final table, some of them are not mentioned during the analytical phase and are only present in the final table.

*the "supporting illustrations" section is not really clear what it is for and what it refers to.

All in all, while I think that this research is highly valuable, some of the parts related to design are not clear as to how they were deduced, it would enrich the research to explain them. 

Again, thank you for your well put research

Round 2

Reviewer 2 Report

The article has improved with significant information and the title now reflects better the subject of the research.